# Advances on Delivery of Cytotoxic Enzymes as Anticancer Agents

**DOI:** 10.3390/molecules27123836

**Published:** 2022-06-14

**Authors:** Akmal M. Asrorov, Bahtiyor Muhitdinov, Bin Tu, Sharafitdin Mirzaakhmedov, Huiyuan Wang, Yongzhuo Huang

**Affiliations:** 1Shanghai Institute of Materia Medica, Chinese Academy of Sciences, Shanghai 201203, China; akmal84a@gmail.com (A.M.A.); muhitdinov.bahtiyor@gmail.com (B.M.); s18-tubin@simm.ac.cn (B.T.); 2Institute of Bioorganic Chemistry, Uzbekistan Academy of Sciences, Tashkent 100125, Uzbekistan; mirzaakhmedov@mail.ru; 3Science Department, Ajou University in Tashkent, Tashkent 100204, Uzbekistan; 4University of Chinese Academy of Sciences, Beijing 100049, China; 5Zhongshan Institute for Drug Discovery, Chinese Academy of Sciences, Zhongshan 528437, China; 6National Medical Products Administration, Key Laboratory for Quality Research and Evaluation of Pharmaceutical Excipients, Shanghai 201203, China

**Keywords:** enzyme drugs, cancer therapy, trichosanthin, gelonin, peroxidase, glucose oxidase, asparaginase, exotoxin, diphtheria toxin

## Abstract

Cancer is one of the most serious human diseases, causing millions of deaths worldwide annually, and, therefore, it is one of the most investigated research disciplines. Developing efficient anticancer tools includes studying the effects of different natural enzymes of plant and microbial origin on tumor cells. The development of various smart delivery systems based on enzyme drugs has been conducted for more than two decades. Some of these delivery systems have been developed to the point that they have reached clinical stages, and a few have even found application in selected cancer treatments. Various biological, chemical, and physical approaches have been utilized to enhance their efficiencies by improving their delivery and targeting. In this paper, we review advanced delivery systems for enzyme drugs for use in cancer therapy. Their structure-based functions, mechanisms of action, fused forms with other peptides in terms of targeting and penetration, and other main results from in vivo and clinical studies of these advanced delivery systems are highlighted.

## 1. Introduction

Enzymes leading to cell damage or death have garnered significant attention since they play key roles in cell metabolism. Cytotoxic enzymes appear to be efficient tools to combat cancer. Ideally, one molecule of a protein toxin would be enough to induce the death of a single cell. However, a number of reasons hamper the utilization of cytotoxic enzymes. Modern approaches to drug delivery and genetic engineering have led to the current stage of their development. The fundamental action of potential cytotoxic enzymes has been highly scrutinized, and studies on these have yielded significant results. Unfortunately, only few of these enzymes have reached clinical stages. Here, we introduce a few mechanisms of action of enzyme drugs currently used in cancer therapies (Figure 1).

### 1.1. Ribosome-Inactivating Proteins

Many of the most efficient toxic enzymes play a significant role in key processes such as protein synthesis; they transduce damage in cell metabolism and amplify it from a single point. Ribosome-inactivating proteins (RIPs) are of these key proteins of plant origin that function to inhibit ribosomal function [1]. Well-known RIPs such as Trichosanthin (TCS), Gelonin (Gel), and others depurinate different sites on ribosomes [2] and, thus, cause cell damage. The well-established anticancer and antiviral activities of these proteins [3] provide advantages over lower molecular weight (MW) anticancer drugs. They are known to inhibit cell viability and tumor cell proliferation and can even induce apoptosis. TCS and Gel also have demonstrated efficiency as cytotoxic agents that cause the damage of different cancer cells. Various smart drug delivery systems have been developed and have been widely investigated to enhance the efficacy of RIPs [2], in particular TCS and Gel, which are type I RIPs with a molecular weight of ~30 kD [4].

### 1.2. Oxidoreductases

Oxidation-reduction state is another important factor for normal metabolism and cell growth. Thus, oxidoreductases, which regulate this, have become key enzymes of high interest. Peroxidases are among the most efficient protein toxins whose role as anticancer agent has been well investigated. These enzymes show direct effects on pH change by consuming one of the main reactive oxygen species (ROS), namely H_2_O_2_. A change in the pH of culture medium or the quality and quantity of substrates derived from their activities can cause oxidative stress and cytotoxicity, thus leading to a disruption in cell metabolism [5]. In a number of studies, as anticancer agents, the cytotoxic roles of horseradish peroxidase (HRP) and glutathione peroxidase (GPX) have been investigated. The role of these enzymes in cancer therapy has also been previously reviewed.

Glucose oxidase (Gox) is another potential toxic protein that has received significant attention in cancer therapy [6]. Both gluconic acid and hydrogen peroxide result from the conversion of glucose, are catalyzed by Gox, and can contribute to anticancer potential. The role of H_2_O_2_ in cancer treatment has been well established, and it has been shown to react with a variety of cellular components [7]. The anticancer role of gluconic acid can be explained by the blockage of citrate uptake, which suppresses tumor growth [8]. A decreased quantity of glucose, which is required in high quantities for tumor growth, can be potentially utilized as another tool to overcome cancer.

### 1.3. Asparaginase

Asparaginase, as an enzyme catalyzing the conversion of L-asparagine into L-aspartic acid and ammonia, is considered to be one the most potentially impactful enzymes that has reached the market. It functions as an asparagine-lowering agent in the blood that deprives acute lymphoblastic leukemia (ALL) cells of asparagine, which results in the inhibition of the growth of these cells [9]. Bacterial asparaginases, available as anticancer agents, have been efficiently used against other types of blood cancers [10]. PEGylation is the most common method in practice used to prolong the efficiency of asparaginase and increase the efficiency of this drug [11].

### 1.4. Bacterial Toxins

Another class of toxic proteins, the effects of which have been widely investigated in cancer therapy, are bacterial toxins. Ribosylation of adenine dinucleotide phosphate (ADP) is a potential mechanism used by several bacterial exotoxins. This activity has been widely investigated in cancer therapy [12]. Pseudomonas exotoxin (PE), an ADP-ribosyltransferase, is among these potential cytotoxic enzyme drugs that have passed clinical studies [13]. Moreover, the clinical efficacy of PE was improved by conjugating with different antibodies or their fragments. In addition, various peptides or protein fragments were also fused to target cancer cells, enhancing the penetrating ability of PEs. Diphtheria toxin (DT) is another potentially useful ribosyltransferase agent, which functions by causing the inactivation of the elongation factor [14]. Both immunoconjugates and fusions of DT were shown to improve its efficiency in terms of targeting and penetrating cancer cells. Additionally, several studies on PE gene delivery have been carried out using biomimetic systems and synthetic polymers, which express the toxin gene in targeted cancer cells, demonstrating their efficient role in cancer therapy.

In this paper, we review several current advances in the drug delivery systems for TCS, Gel, HRP, peroxidases, asparaginase, PE, and DT.

## 2. Trichosanthin

TCS is a widely investigated ribosome-inactivating protein. A number of investigations have reported its antiviral activity towards Hepatitis B virus [15] and its inhibitory effect on HBsAg (the surface antigen), HBeAg (viral protein of hepatitis), and Herpes simplex virus 1 (HSV-1) [16]. Moreover, HIV infectivity was found to be weakened by TCS-enrichment [17]. Zhao et al. further demonstrated the ability of TCS to penetrate into HIV type 1 particles [18]. The anti-HIV property of TCS was found to be associated with a significant enhancement in the activation of chemokine receptors expressed in HEK293 (human embryonic kidney) cells [19].

The above-mentioned results indicate the potential of TCS. Moreover, TCS has been shown to have even greater significance in cancer therapy. Earlier studies revealed that TCS could induce apoptosis in MCG803 human stomach adenocarcinoma cells [20]. Cell viability levels impacted by TCS were both concentration- and time-dependent. The protein expression levels of p53, which triggers apoptosis [21], were highly reduced, and p21, a cyclin-dependent kinase inhibitor [22], was concurrently upregulated in these studies. The suppression of CNE1 carcinoma epithelioid cells and CNE2 poorly differentiated nasopharyngeal carcinoma cells by TCS was attributed to induced apoptosis and partial suppression of telomerase activity [23]. TCS was more efficient at inhibiting CNE1 tumor growth. TCS was concluded to induce tumor cell apoptosis. Studies on the effects of TCS on HL60 leukemia cells showed that caspase-3 and -8 were activated in both the intrinsic (mitochondrial) and extrinsic (endoplasmic reticulum) apoptosis pathways [24]. The induction of the caspase-9-dependent and caspase-4-dependent pathways by TCS was also found to lead to apoptosis. Smac, an activator of caspases, was found to be upregulated in TCS-treated CaSki cervical cancer cells [25]. Smac has been proposed to act reversibly to modulate chemoresistance to TCS and inhibit resistance. TCS was revealed to have the ability to reduce the level of DNA methyltransferase 1 genes and proteins in Hela and CaSki cervical cancer cells as well and to restore the gene expression levels of methylation-silenced tumor suppressors [26]. Synergic effects of interleukin-2 (IL-2) on TCS were also studied in PC3 (prostate cancer) cells [27] (Table 1). Significant differences in tumor volume and weight were found between TCS and TCS/IL-2 treatments in mice bearing PC3 cells. Moreover, TCS highly reduced procaspase-8 and caused significantly increased levels of caspase-8, an apoptosis marker in cancer [28]. Depending on the tumor cell type, the pathways leading to apoptosis may differ [29]. Additionally, changes in ROS levels may also cause the death of tumor cells.

The TCS uptake levels by various cells significantly differ, even among cancer cells. It has been reported that the entry process of TCS in JAR cells happens through specific receptors through rapid accumulation [30]. Slow and non-specific penetration in H35 cells has been suggested to result from direct diffusion across the membrane of H35 cells. Low-density lipoprotein receptor-related protein 1 (LRP1) has been reported to be responsible for the binding of TCS [31]. The cell viability levels of JAR and Bewo human choriocarcinoma cells were inhibited to <20% at a 20–100 µg/mL concentration of TCS, compared to controls. However, in HeLa (ovarian cancer) cells, the effects were at least 2-fold lower, which was explained by the low expression level of LRP1 on the cell surface. G protein-coupled receptor 5 (LGR5), which is rich in leucine, was reported to be targeted by TCS in U87 glioblastoma cells [32]. Suppressed levels of LGR5 as a result of TCS treatment was accompanied by reduced protein levels of β-catenin, c-myc, pGSK-3βSer9, and cyclin D1, which are all involved in cancer progression [33].

Studies on the effects of TCS on survival in immunocompetent and nude mice bearing Lewis lung cancer cells revealed the involvement of host immune cells in tumor eradication [34]. TCS was found to enhance effector T cell and interferon γ levels. CD4+CD25+ T cells, a subtype of T lymphocytes, were demonstrated to be significantly enhanced by TCS treatment, and these TCS-enhanced cells possessed higher suppressive activity as effector T cells in vitro [35]. However, this effect was dependent on cytokine secretion and cell–cell contacts. TCS was found to increase cytosolic Ca^+2^, which further suppressed protein kinase C (PKC) rather than protein kinase A in HeLa cells [36]. The reduction in cAMP levels induced by TCS was observed to be time- and concentration-dependent. Earlier studies revealed that the suppression of HeLa cell proliferation by TCS was regulated by the PKC/MAPK pathway [37]. The cell death of MCF-7 (estrogen-dependent) and MDA-MB-231 (estrogen-independent) human breast cells, resulting from TCS influence, was linked with the activation of both the caspase-8 and -9 pathways [38]. Their dose-dependent activation followed the enhancement of caspase-3 and further PARP cleavage.

Other mechanisms of TCS leading to tumor inhibition and cell death were reported to use the NO and caspase-4 pathways [39]. To date, highly efficient effects on more than 20 different cells have been detected [2,39]. Further studies on TCS are directed at using TCS efficiently and have been mainly devoted to delivering TCS to tumor cells and keeping it intact during circulation in the blood.

Various approaches to enhancing the delivery and targeting of TCS to tumor cells have been developed to date. In this regard, both synthetic and natural higher and lower molecular weight compounds have been used. In this section, we review these drug delivery systems and their outcomes.

Developing smart drug delivery systems (DDSs) utilizes various approaches to overcome challenges in targeting and delivering. In this aspect, the inclusion of cell-penetrating peptides (CPPs) has greatly contributed to the development of intracellular delivery systems [40]. Providing a stable connection between protein drugs and CPPs has been realized by utilizing recombinant proteins, covalent chemical bonding, noncovalent interactions, or surface modifications. Thus, different strategies have been developed based on enzymatic cleavage, pH triggering, or light-dependent changes. One smart DDS for delivery of TCS into tumor cells has included lower molecular weight protamines (LMWPs), which mainly consist of arginine (VSRRRRRRGGRRRR), matrix metalloproteinase-2 (MMP-2) substrate (MSP) peptide, and PEG [41] (Table 1). The role of MMP-2 substrate is to allow for specific cleavage at tumor sites. Thus, the PEG cover would be removed from internalized TCS. The remaining fusion of TCS with LMWP further demonstrated a higher efficiency. This system resulted in the highest tumor size inhibition, which was at least 2-fold more efficient than TCS or TCS-LMWP that was not coated with PEG. The differences in tumor weight were even greater, which highlights the efficiency of LMPW and PEG. Further development included studying the effects of PEGylated TCS or TCS-LMWP-MSP together with paclitaxel (PTX) on PTX-resistant A549/T NSCLC cells [42]. A general mechanism of obtaining these conjugates is illustrated in Figure 2. The developed approach revealed the synergistic effects of TCS-LMWP-MSP and PTX in liposomes in vivo.

A similar approach was used to deliver TCS to glioma cells [43]. In this work, the fusion of TCS, LMWP, and MMP-2 substrate peptide (PLGLAG) was further linked with Lactoferrin (TCS-LMWP-MSP-Lf) to pass the blood–brain barrier. The Lf-mediated passage was then evaluated with an in vitro model. A high uptake level of the developed protein fusion was demonstrated by fluorescent imaging in GL261 cancer cells. A similar therapeutic effect on tumor volume in nude mice was detected, with a 2-fold lower concentration of fused TCS. Significant differences in tumor volume were found in a C57 mouse model. Compared to recombinant TCS, the fused rTCS led to a several-fold lower tumor volume after 24 days. Moreover, the antiglioma activity of TCS was enhanced, and the immunogenicity was shown to be lowered.

Another trial towards improving the efficiency of TCS included albendazole (ABZ) and silver nanoparticles (Ag NPs) [44]. This system was developed by conjugating TCS-LMWP with ABZ covered with albumin and Ag NPs. The developed nanosystem had demonstrated high antitumor efficiency resulting from various mechanisms, including ribosome inactivation by TCS, skeleton dysfunction by ABZ, and lowered mitochondrial membrane potential by Ag NPs. Altogether, these actions led to a total apoptosis rate >60% and >45% in A549/T and HCT8/ADR cells, respectively (Table 1). Albumin, as a rich source of negatively charged amino acids (glutamic acid and aspartic acid), provided an electrostatic interaction with LMWP-conjugated TCS. The main role of albumin, however, in the developed smart DDS, was targeting cancer cells, as many solid tumors are known to promote higher albumin uptake and overexpress albumin-binding proteins [45]. This kind of nanodelivery system using combined drugs could serve to overcome drug-resistant tumors and prevent metastasis in a variety of tumor types.molecules-27-03836-t001_Table 1Table 1Summary of recent studies on TCS-based nanomaterials for cancer therapy.FormulationResultsIC50 ValuesRefs.Combination of TCS with Interleukin-2Combination with IL-2 resulted in synergistic effect on PC3 prostate cancer cells in vivo.50.6 µg/mL (PC3)[27]Protection of TCS-LMWP-MSP with PEG The inclusion of MSP (MMP-2 substrate) between TCS-LMWP and PEG inhibited HT1080 tumor volume and weight twice more efficiently.0.13 µM (HT1080)[41]Protection of TCS-LMWP-MSP with PEGNo significant changes were observed in A549/T cells in vivo, but the combination with PTX in liposome totally inhibited the tumor volume.1.6 µM/mL (A549/T)[42]Fusion of TCS-CPP-MSP with Lactoferrin (LF)The linkage of LF with TCS-CPP-MSP lowered the IC50 value and enhanced GL261 tumor inhibition significantly.0.37 µM (GL261)[43]Co-delivery of TCS and albendazole by self-assembly via BSA and Ag NPsNumber of metastatic nodules dramatically reduced. Tumor volume and weight in mice bearing multidrug-resistant A-549/T cell greatly inhibited, and thus concluded to prevent lung metastasis.<0.1 µg/mL (A549/T and HCT8/ADR cells)[44]Development of recombinant ABD-PTN-TAT-TCS proteinRecombinant TCS capable of binding albumin greatly inhibited tumor in mice bearing 4T1 cells.1.7 µM (4T1)[46]

A further genetically engineered prodrug form, termed the Smart Hitchhike via the Endogenous Albumin-Trichosanthin Hinge (SHEATH), has been developed based on these refinements [46]. The constructed plasmid, transformed to *E. coli*, contained an albumin binding domain (ABD), a protease substrate (PTN), a TAT peptide (YGRKKRRQRRR), and TCS (Table 1). The obtained SHEATH system endowed affinity to albumin, and in mice bearing 4T1 orthotopic xenografts, it inhibited both the tumor weight and volume 3-fold more efficiently as compared to free TCS. Interestingly, the tumor-inhibiting efficacy of TAT-excluded ABD-PTN-TCS conjugates was slightly better than that of ABD-PTN-TAT-TCS. Both ABD-PTN-TCS and ABD-PTN-TAT-TCS efficiently enhanced apoptosis levels in tumors.

## 3. Gelonin

The mature Gel protein of plant origin is composed of 258 amino acids and has 21 lysine residues [47]. It possesses 33% sequence similarity to TCS and the ricin A chain. Different recombinant forms of Gel have been constructed and developed to target different types of tumors [2,48]. Early research evidence on the extraction and purification of Gel came from *Gelonium multiflorum* [49,50]. The antibody targeted, triggered, electrically modified prodrug-type strategy (ATTEMPTS) system is expected to very efficiently deliver protein drugs based on antibody-associated cancer cell targeting. Non-covalent bonding between CPPs and heparin provides interactions that are stable during blood circulation and prone to disaggregation with the addition of protamine [51,52]. Earlier trials to deliver CPP-conjugated Gel using a T84.66 murine anti-CEA monoclonal antibody demonstrated the efficiency of this strategy [51] (Table 2). Electrostatic interaction between antibodies and toxins was provided by conjugating heparin to T84.66 via a thioester bond. The antibody-targeted CPP-Gel system accumulated toxin in tumor tissues at levels 43-fold higher compared to free CPP-Gel.

Electrostatic interactions formed between CPP, conjugated with Gel, and Heparin, conjugated with T84.86, resulted in significantly lower tumor size compared to toxin-CPP conjugation (TAT-Gel) [52]. The addition of protamine (Pro) resulted in a TAT-Gel/T84.86-Hep + Pro system that improved targeting and delivery efficiency. In vivo experiments revealed a 3-fold inhibition of tumor in mice after treatment with this ATTEMPTS system.

Protamine was found to be important for TAT-Gel fusions that were electrostatically attracted to T84.66-Hep [53]. These heparin-included ATTEMPT systems could be efficiently utilized for cancer therapy, but the anti-coagulation property may cause unexpected problems at higher doses.

Another approach to improve the efficacy of Gel employed genetic engineering of its fusion with the TAT peptide [54,55]. A TAT-Gel fusion gene was further constructed with a protease inhibitor (TEVp), a 6xHis, and a thioredoxin motif, imbuing this fusion with affinity to nickel charged nitrilotriacetic acid resin (TRX). This TRX-6xHis-TEVp-TAT-Gel protein fusion, when transformed into *E. coli*, was efficiently expressed. The TRX-6xHis tagged site was used to purify the fusion in the initial stages through elution with imidazole. Further purification was carried out using TEV protease. The obtained TAT-Gel fusion was covered by Heparin (Hep), and the formed electrostatic interaction provided high stability that could be destroyed by protamine addition. This TAT-Gel/Hep system alone did not have effects on LS174T (colorectal cancer) cells, but with the addition of Protamine (Pro) (termed TAT-Gel/Hep + Pro), it reduced the cell viability level to that of TAT-Gel. The developed TAT-Gel/Hep + Pro system reduced the tumor volume to a third of the level in mice bearing LS174T cells. It has been reported that the cytotoxicity of the TAT-Gel fusion in different cancer cells was at least 53-fold higher compared to Gel alone.

A genetic approach for improving the anticancer properties of Gel focused on fusing Gel with a lower MW human vascular epithelial growth factor (VEGF_121_) [56]. This genetically engineered Gel fusion was linked to VEGF by a flexible GGGGS sequence (VEGF_121_/Gel). This VEGF_121_/Gel fusion was shown to lead to significantly reduced tumor volume in mice bearing A375M cells, whereas rGel itself did not affect tumor volume. Furthermore, VEGF_121_/Gel was combined with photochemical internalization (VEGF_121_/Gel-PCI) to improve its therapeutic efficacy, and its effect on VEGFR1 and VEGFR2 receptors was determined [57] (Table 2). This study confirmed the cancer targeting ability of this molecule via interaction with VEGFR receptors and direct tumor cell targeting, independent of receptors. MTT assay data verified that treatment with VEGF-Gel-PCI reduced the cell viability degree by 4- and 10-fold at 48 h, when CT26, CT25, and 4T1 cells were exposed to light for 80 and 120 min, respectively.

In another genetic approach, fusion of melittin (GIGAVLKVLTTGLPALISWIKRKRQQ), a pore-forming peptide toxin, with Gel resulted in increased cell uptake levels [58]. This Gel fusion was obtained by chemical conjugation (cGel-Mel) and by genetic engineering (rGel-Mel) (Figure 3) and was shown to lower the IC_50_ 32- and 10-fold, respectively. Compared to a mixture of Gel and mellitin, the cell viability levels of various cancer cells (HeLa, CT26, 9L, LS174T, and U87MG cells) lowered by 2- and 3-fold at the optimum concentration (10^−6^ M) of rGel-Mel and cGel-Mel, respectively. Similar results were observed with non-cancer MDCK cells. cGel-Mel was shown to have better antitumor effects in terms of inhibition of both tumor volume and size by more than 3-fold.

Flexible short peptides, composed of GGGGS, were also used to construct a Gel fusion with epidermal growth factor (EGF) [59], the receptor of which (EGFR) is overexpressed on the surface of tumor cells [60]. The Gel-EGF fusion and Gel-EGF combined with PCI resulted in different effects on cell viability, with PCI enhancing the prodrug efficiency [59]. The formation of a dimer of rGel/EGF and EGFR and cross-phosphorylation led to internalization. Furthermore, the TPCS2a photosensitizer entered cells following the rGel/EGF-EGFR complex and produced ROS (^1^O_2_), which caused the release of rGel-EGF into the cytosol. In A-431 human epidermoid carcinoma and WiDr colon adenocarcinoma cell lines, Gel-EGF-PCI treatment significantly suppressed the cellular viability levels compared to Gel-PCI. However, adverse effects were observed in MES-SA human sarcoma and MDA-MB-435 melanoma cells, which could be explained by the barely detectable EGFR expression levels. A significant reduction in tumor volume compared to untreated controls was observed under the influence of Gel-EGF-PCI. Gel monotherapy or Gel-PCI did not result in significant changes in tumor volume. Thus, the fusion with EGF was demonstrated to be an effective therapeutic means when combined with PCI. Additionally, using PCI was shown to increase the efficacy of Gel or its fusions in other cancer cells [61,62]. Enhancements were demonstrated in uterus cancer, breast cancer, colon cancer, ovarian cancer, sarcoma, bladder cancer, glioma, skin cancer, and lung cancer [63].

Another genetic recombinant approach fused Gel with the anti-insulin-like growth factor-1 receptor (IGF-1R), a 58 amino acid long peptide that can specifically bind to receptors, overexpressed in cancer cells [64]. This fusion had no significant impact on the N-glycosidase activity of Gel. This genetically fused protein greatly suppressed the cell viability (~10-fold) of U87 MG and U251 MG glioma cancer cells (10^−6^ M). This conjugate of Gel with IGF-1R suppressed cell viability of U87 MG and U251 cells 5- and 6-fold, respectively, in this study. However, no significant changes were detected in lymph node carcinoma of the prostate (LNCaP) cells in either case. The fusion of Gel with single, double, and triple F3 peptides (F3-Gel, 2F3-Gel, and 3F3-Gel) was also demonstrated to significantly increase Gel efficiency [65]. 2F3-Gel and 3F3-Gel fusions suppressed the cell viability levels of LNCaP, PC-3, and DU-145 prostate cancer cells by at least 6-fold and were found to be more efficient than F3-Gel. The uptake levels of F3-Gel by U87 MG and 9L glioma cells were 2.7 and 3.2 times greater than that of Gel, which indicates the significant role of F3-combination [66] (Table 2). The viability levels of U87, 9L, LNCaP, and Hela cells were significantly lowered by F3-Gel at the optimum concentration (10^−6^–10^−7^ M). Between the cell viability and uptake levels of Gel and F3-Gel by non-cancer HEK 293 and SVGp12 cells, respectively, no significant changes were detected. The tumor volumes of U87 MG xenografts were also greatly decreased by F3-Gel to as much as ~40% of the mean value of that resulting from treatment with Gel.

The fusion of brain cancer homing peptide, chlorotoxin, with Gel was also found to be selectively internalized in U-87 MG glioma cells [67]. Compared to Gel and Gel + chlorotoxin samples, the cell viability levels of U87 MG and 9L glioma cells were suppressed 10- and 8-fold by this chlorotoxin-Gel fusion, respectively (10^−6^ M). The fused Gel possessed equipotent N-glycosidase activity and significantly inhibited tumor growth compared to unfused Gel, which led to a 4-fold lower volume relative to that resulting from treatment with PBS.

To target lymphocytes during chronic lymphocytic leukemia, Gel was fused with B lymphocyte stimulator (BLyS) [68]. This genetically constructed Gel-BLyS could specifically bind to cell surface receptors. The fused Gel could internalize in leukemic lymphocytes with an IC_50_ < 3 nM, which was only equaled by 5000 nM Gel. Furthermore, Gel-BLyS was shown to lead to downregulation of interleukin-6 receptor, a protein that is highly expressed in diffuse large B cell lymphoma (DLBCL) and impacts the STAT3 targets c-Myc, p21, Mcl-1, and Bcl-xL [69]. Moreover, STAT3 phosphorylation levels and STAT3-DNA binding activity were found to be inhibited by Gel-BLyS treatment. The growth of B cell-like ABC-DLBCL cell lines (OCI-Ly3 and OCI-Ly10) was totally inhibited at a 10^6^ pM concentration of Gel-BLyS, while SUDHL-4 and SUDHL-6 cells were unaffected by this treatment. Thus, Gel-BLyS was suggested to be an efficient candidate for treating ABC-subtypes of DLBCL. Gel-BLyS was also found to downregulate the levels of bcl-xL, MCL-1, which are the targets of nuclear factor kB (NF-kB) [70]. The cytotoxic effects of Gel-BLyS were also suggested to be mediated via BLyS receptors, which induced apoptosis by activating caspase-3 and cleaving poly (ADP-ribose) polymerase [71]. Thus, the potential role of BLyS in targeting BLyS receptor-overexpressing B cells was explained. Main results of Gel-based nanomaterials are summarized in Table 2.molecules-27-03836-t002_Table 2Table 2Summary of recent studies on Gel-based nanomaterials for cancer therapy.FormulationResultsIC50 ValuesRefs.ATTEMPTS rGel-TAT + T84.66-Hep + ProtamineTargeting by T84.66-Hep enhanced the drug tumor accumulation 43-fold compared to free recombinant form of Gel-TAT.29.2 nM in LS174T cells for Gel-TAT.[51]ATTEMPTS rGel-TAT + T84.66-Hep + ProtamineDeveloped ATTEMPTS system efficiently inhibited tumor volume in mice bearing LS174T cells. Targeting by T84.66 antibody and release by protamine and heparin resulted in two-fold efficiency in vivo.79 nM (CT26), 68 nM (LS174T), 61 nM (9L), 84 nM (PC3).[52,53] Combination of Gel-TAT with Hep + protamineSimilar IC50 values were observed with non cancer MDCK and 293 HEK cells. Gel-TAT and Gel-TAT/Hep + protamine treatment efficiently inhibited LS174T tumor volume with insignificant changes. 72 nM (LS174T), 46 nM (U87 MG), 58 nM (9L), 66 nM (Hela).[54]Combined action of fusion of Gel-VEGF121 and PCIFusing with VEGF121 enhanced the Gel cytotoxicity in PAE/VEGFR-2 and PAE/VEGFR-1 cells more than 400- and 250-fold.44 pM (PAE/VEGFR-2), 26 pM, (PAE/VEGFR-1).[57]Construction of recombinant Gel fusion with EGFR targeting sequenceDespite great in vitro activity, no significant changes were observed in SCC-026 tumor volume in mice. The integration of PCI inhibited the tumor volume by ~45% compared to untreated control.60 nM (SCC-026), 3.7 nM (SCC-040), 19 nm (SCC-074).[59]Construction of recombinant Gel affibody with IGF-1R (Gel-IAFF)Cell viability levels of U87 MG and U251 MG cells enhanced by more than 10-fold by the inclusion IGF-1R sequence. No significant changes were found 293T and LNCaP cells.0.18 nM (U87 MG), 0.14 nM (U251MG).[64]Fusing Gel with F3 peptides: F3-Gel, 2F3-Gel, and 3F3-Gel2F3-Gel and 3F3-Gel fusions inhibited LNCaP tumor volume twice more efficiently compared to free Gel. 2F3-Gel and 3F3-Gel revealed significantly lower IC50 values in HEK cells.63 nM (LNCaP), 99 nM (PC3), 73 nM (DU145) for 2F3-Gel.[65]Construction of recombinant Gel fusion F3 peptideFusing with F3 peptide lowered the IC50 value of Gel in HeLa LNCaP, 9L, and U87 MG cells at least 6-fold. The fusion inhibited U87 MG tumor volume 5-fold more efficiently than Gel.0.34 µM (HeLa), 0.41 µM (LNCaP), 0.39 (9L), 0.33 (U87 MG).[66]Construction of a recombinant Gel fusion with chlorotoxin (Gel-CLTX)Obtained recombinant fusion increased Gel toxicity ~20-fold to U87 MG and 9L cells. No significant changes were observed in the toxicity level to non cancer 293 HEK and SVG p12 cells.180 nM in U87MG and 9L cells.[67]Fusion of Gel with BLyS that target BLyS receptorsDirect correlation was established between BAFF-R level and sensitivity to BLyS-Gel. No correlation was found with TACI protein.5–50 nm in ABC DLBCL cell lines[69,71]Fusion of Gel with B cell lymphocyte stimulator (BLyS) that target BLyS receptorsGenerated BLyS-Gel fusion totally inhibited and reduced DLBCL xenograft tumor in mice. The fusion showed great targeting index in cells that overexpress BLyS receptors.7 pM (OCI-Ly10)8 pM (OCI-Ly3)0.1 nM (SUDHL-4)5 nM (SUDHL-6)[70,71]

Anti-Her2/neu conjunction with Gel has been developed using human single chain antibodies (C6.5) [72]. Among the three different types of linkers, GGGGS, TRHRQPRGWEQL (Fpe), and AGNRVRRSVG (Fdt), utilized for the construction of Gel fusion, GGGGS has been demonstrated to have the highest efficiency. The tumor volume of nude mice bearing SKOV3 cells was efficiently inhibited by C6.5-GGGGS-Gel. This suppressed the tumor volume 3-fold compared to Gel alone.

Liposomes are an alternative approach for toxin delivery and have been efficiently used to deliver Gel to cancer cells [73]. Co-encapsulation of Gel and listeriolysin, a pore-forming protein, was shown to enable delivery into B16 melanoma cells. To this end, pH-sensitive liposomes were prepared from 1,2-dioleoyl-sn-glycero-3-phosphoethanolamine (PE) and cholesteryl hemisuccinate. The developed delivery system enhanced the Gel in vivo efficiency in mice bearing B16-F10.

Another approach to increase the Gel efficiency utilized chemically bound LMWP (VSRRRRRRGGRRRR) linked via NHS-PEG-PDP (2 kDa) [74]. The developed chemical conjugate reduced the viability levels of CT26, LS174T, and PC3 cells by at least 3-fold, compared to free Gel. Similar results were observed with genetically engineered Gel-LMWP. Compared to Gel, Gel-LMWP inhibited the tumor volume by over 6-fold in mice bearing CT26 cells.

## 4. Peroxidase

Oxidation-reduction changes are an efficient mechanism used to treat cancer cells. HRP and GPX play important roles in maintaining cellular processes. Studies have been carried out to improve the efficacies of these enzymes in cancer therapeutics. In this section, we review the delivery aspects of peroxidase enzymes to cancer cells.

HRP belongs to the class III family of peroxidases. Indoles, aromatic phenols, and sulfonates, together with H_2_O_2_, are considered to be its substrates [75]. HRP has been demonstrated to have high stability and activity at 37 °C and a neutral pH [76]. H_2_O_2_ plays the role of a mediator during apoptosis resulting from indole-3-acetic acid (IAA)/HRP activity [77]. Cell viability levels of G361 human melanoma cells were found to be reduced by ~10% by the tandem application of IAA (500 µM) and HRP (1.2 µg/mL) for 8 h. The addition of catalase (U/mL) and NADPH (50 µM) led to a reduction of <60% and <20%, respectively. The apoptotic pathway was indicated to be supported with H_2_O_2_ generated by IAA/HRP. Additionally, the rate constants for the removal of H_2_O_2_ were found to differ in normal and tumor cells; kcell = 5.5 × 10^–12^ s^−1^ cell^−1^ L was found for normal cells, and kcell = 3.1 × 10^–12^ s^−1^ cell^−1^ L was found for tumor cells [78]. The kcell rate constants’ average mean values of MIA PaCa-2 (pancreatic cancer), A375 (melanoma), and MB231 (breast cancer) cells were found to be the lowest at 1.1, 0.65, and 1, respectively. On average, this parameter was found to be 2-fold lower in cancer cells than in normal cells and correlated with ascorbate concentration.

GPX is an enzyme that reduces H_2_O_2_ to water using two glutathione that form a glutathione dimer [79]. Cancer stem cells contain lower amounts of ROS that are not compatible with free radical scavenging systems [80,81]. In mammalian cells, glutathione is present at millimolar levels and acts as a significant regulator of redox processes [82]. Different types of breast cancer cells (MCF-7, T-47D, MDA-MD-231, Hs578T, and BT549) express several-fold lower GPX1 levels compared to normal cells. Their total activity was also determined to be significantly lower [83]. Recently, novel approaches have been developed to target glutathione-linked antioxidative pathways in cancer cells, which include DNA synthesis/repair, antioxidant defense, protein synthesis, and cell division and proliferation, as well as post-translational modifications or ferroptosis [82].

Several trials have been carried out to deliver oxidoreductase enzymes to cancer cells. In a trial of gene-directed delivery of HRP, human T24 bladder carcinoma cells were sensitized with IAA [84]. The developed gene-directed delivery of HRP was found to cross cell membranes. Within 2 h, HRP-IAA treatment enhanced T24 cell viability levels by over 9-fold compared to HRP or IAA treatments. In normoxic conditions, concentration-dependent changes of IAA quantity were observed after treatment with HRP prodrug forms.

A-1,6-mannosyltransfererase-knocked-out *Pichia pastoris* strains were used to produce recombinant HRP to target cancer cells with reduced surface glycosylation [85]. In MDA-MB-231 breast carcinoma cells, the obtained HRP could reduce the IAA concentration to 1 mM/L, which equaled ~20%. The enzyme activity was similar to that of commercial HRP. In T24 bladder carcinoma cells, the enzyme activity was even greater than that of commercial HRP. Glyco-engineered recombinant HRP revealed high cytotoxic effects (IC_50_ 0.115 mM/L; T24 cells) at lower amounts of IAA, which were 4-fold lower than that of plant HRP. The authors specifically proposed producing recombinant HRP and utilizing it in antibody-directed enzyme prodrug therapy in combination with IAA (Figure 4).

Chitosan and chitosan-PEG nanoparticles were demonstrated to immobilize HRP and thus influence its activity [86]. At a pH of 7, no significant differences between the activities of free HRP and HRP immobilized on chitosan (HRP/CS) and chitosan-PEG (HRP/CS-PEG) were observed. Immobilization in CS NPs significantly enhanced the enzyme activity at pH 6, 8, and 9. Kinetic parameters for free and immobilized enzymes were determined to be concentration dependent.

CPP-modification of HRP was also shown to enhance its cell permeability [87]. The enzymatic activity of HRP-CPP conjugate linked using an anthroquinone moiety was similar to that of free HRP. The cell permeability and enzymatic activity were maintained in the absence of photo-irradiation.

## 5. Glucose Oxidase

GOx is structured from a dimeric glycoprotein consisting of two identical polypeptide chain subunits that are covalently linked together by disulfide bonds. These GOx protein subunits are glycosylated with mannose-rich carbohydrate chains and contain non-covalently bonded flavin adenine dinucleotides as co-factors at their conserved active sites. GOx possesses a MW ranging from 130 to 175 kD and has overall dimensions of 6.0 × 5.2 × 7.7 nm [88,89]. GOx, using molecular oxygen as an electron acceptor, catalyzes the oxidation of β-D-glucose to H_2_O_2_ and D-glucono-1,5-lactone, which is further hydrolyzed to gluconic acid. This enzyme has been extensively studied for developing glucose sensors [90,91] and glucose-mediated insulin delivery systems [92,93,94]. In recent years, GOx has attracted growing interest for use in cancer starvation therapy. This approach is mainly related to cutting off the nutrition source for cancer by employing this enzyme to reduce glucose levels, a crucial nutrition source for cancer cells [95] Tumor cells tend to uptake and consume more glucose than normal cells, giving them faster proliferation (Warburg effect), as glycolysis is considered the main energy resource for tumor progression [96]. GOx can oxidize intracellular glucose in the presence of oxygen to produce hydrogen H_2_O_2_ and gluconic acid, thus cutting off the nutrition source of cancer cells and consequently inhibiting their proliferation [97] (Table 3). In addition to this glucose-dependent cancer starvation, glucose oxidation, by consuming oxygen and producing gluconic acid, induces hypoxic and acidic conditions in tumor cells and the tumor microenvironment. The produced H_2_O_2_ can incite cancer oxidative stress and generate cytotoxic hydroxyl radicals, leading to cancer cell death [6]. However, GOx-induced biochemical reactions and derived changes can disrupt immune evasion mechanisms in tumor cells by reinforcing antigen production. In light of the above characteristics, enhanced antitumor effects can be expected from GOx-based therapeutics. In addition, the GOx-induced versatile biochemical changes in tumor cells and the tumor microenvironment are compatible with other therapeutic strategies such as chemotherapy, immunotherapy, chemodynamic therapy, phototherapy, and gas therapy to achieve multimodal integrated therapy and overcome single mode therapy-related issues [98]. Despite promising cancer therapeutic characteristics, GOx is highly prone to degradation in biological conditions and suffers from the problems of a short in vivo half-life, immunogenicity, and systemic toxicity [97,99] In addition, direct application causes this enzyme’s unspecific spread since there is no cancer-specific transportation pathway in humans. Therefore, some nanostructures have been developed for tumor-specific delivery of GOx, which function by protecting and preserving GOx activity and are more effective for probing with other integrated multimodal approaches to enhance cancer starvation therapy. 

GOx is one of the enzymes that can be efficiently used for cancer treatment. It catalyzes the conversion of glucose and oxygen into gluconic acid and hydrogen peroxide. Anticancer mechanisms of actions of GOx include the reduction/suppression of glucose, induction of hypoxia, increase in acidity resulting from gluconic acid, and highly enhanced levels of H_2_O_2_, which directly results in oxidative stress [98]. Glucose deprivation in cancer cells will cause starvation, which can be combined with oxygen shortage. On the other hand, H_2_O_2_ formation supports cancer cell death. Enzyme delivery of GOx to tumor tissue/cancer cells still faces many challenges. Its short in vivo half-life and systemic toxicity effects require its protection during circulation in the bloodstream. Despite this, various approaches have been used to deliver GOx to tumor microenvironment/cancer cells. Hollow mesoporous structures, such as silica NPs, metal-organic systems, magnetic NPs, and polymers that can exert multimodal synergistic effects, have been analyzed for the delivery of GOx to cancer cells [97]. To integrate GOx carrying systems, physical interactions were more often used rather than chemical conjugations [95].

The enhancement of OH radical formation was shown to be maintained by developing Fe_3_O_4_/GOx-polypyrrole (PPY)-based nanocomposites, which were further integrated with photothermal conversion tools (Figure 5) [100]. OH radical formation from H_2_O_2_ was shown to be supported by Fe_3_O_4_ and PPY + NIR treatment. Fe_3_O_4_@PPY@GOx-NIR I (808 nm) and Fe_3_O_4_@PPY@GOx-NIR II (1064 nm) treatments reduced the 4T1 (breast cancer) cell viability several-fold compared to treatments with NIR I, NIR II, Fe_3_O_4_@PPY + NIR I, or Fe_3_O_4_@PPY + NIR II individually. In vivo experiments carried out on mice bearing 4T1 cells demonstrated a complete absence of tumors after treatment with Fe_3_O_4_@PPY@GOx following NIR I and NIR II treatment. The effects of photothermal conversion with NIR II were found to be greater.

Another chemodynamic therapy approach utilized GOx in the zeolitik imidazole framework (ZIF) that was coated with Fe^+3^ ions and tannic acid (TA) (MPN) [101]. TA triggered the reduction in Fe^+3^ into Fe^+2^, thus contributing to OH radical formation. The cell viability levels of ATP-overexpressing 4T1 cells, in the presence and absence of glucose, decreased significantly after GOx@ZIF@MPN treatment, compared to treatment with ZIF@MPN. The tumor volume in mice bearing 4T1 cells was reduced by this developed nanosystem, whereas no significant changes were detected between PBS and ZIF@MPN samples, in which the initial tumor volume increased 6- and 7-fold for 14 days. GOx@ZIF particles inhibited the tumor volume by ~50%. In another work, ZIF was used to integrate GOx and Zn^+2^, which was further coated with 4-mercaptobenzonitrile (MBN)-modified Ag NPs (AgNPs@MBN) [102] (Table 3). At 50 µg/mL concentration, the fabricated nanosystem with an average diameter of ~100 nm could completely inhibit HeLa tumor size in mice, and this equaled half of the volume of that resulting from GOx@ZIF treatment. Ag^+^ accumulated in tumors several-fold higher than in other tissues, but Zn^+2^ accumulated in liver cells more than in tumors. GOx@ZIF@AgNPs@MBN led to no significant quantitative changes in RBC, HGB, HCT, or MCH compared to PBS treatment.

Another approach of combined actions, namely, starving, photodynamic, and photothermal therapies, resulted in the integration of poly(γ-glutamic acid) (γ-PGA), enzyme drug, and Mn, Cu-doped carbon dots (γ-PGA@GOx@Mn, Cu-CDs) [103]. Laser treatment was proven to exert photodynamic and photothermal effects. In all experimental groups, the developed nanosystem had an average diameter of 80 nm and led to total tumor inhibition under 1.5 W/cm^2^ laser treatment.

In another case, glucose was chemically conjugated to the surface of nanosize MnO_2_ (MNS-GOx) modified using melanin [104]. MnO_2_ can catalyze decomposition reactions to form O_2_ that can support glucose oxidation, thus exerting synergistic effects. The effects of the fabricated MNS-GOx nanodelivery particles (with an average diameter of ~70 nm) on A375 human melanoma cells were not remarkable. However, MNS-GOx + laser treatment increased its efficiency 4-fold. MNS-GOx + laser treatment completely dispersed the initial 6 mm tumors in mice, whereas MNS-GOx and MNS + laser treatments resulted in tumor inhibition by 45% and 85%, respectively. Moreover, the survival rate was 75% at 28 days.

Loading GOx inside an alginate/chitosan-based microemulsion resulted in improved biological properties [105]. Spherical particles with an average diameter of 138 µm could conserve 90% of GOx enzyme activity during lyophilization and storage at −20 °C for 30 days. Lower pH values (3–4) resulted in higher loading capacities, as high as ~10%. The addition of chitosan was also found to significantly reduce loading efficiencies. Freeze-drying was found to still be compatible with enzymatic activity, and the developed emulsion has been suggested as a protecting system for GOx. Further analysis, carried out to enhance the physicochemical parameters of these particles, demonstrated higher production of H_2_O_2_ as a result of their smaller size, ~4 µm [106]. MDR breast cancer cells were found to be more damaged than WT cells. Compared to higher sized particles (20–120 µm), smaller micro-sized particles exerted greater effects on cell viability, relative cell growth, lactate dehydrogenase leakage, and H_2_O_2_ formation. However, these parameters still need more improvement to reach the levels of activity seen using free enzymes, and a correlation between cell viability and membrane damage demonstrates the significance of the developed micro-sized GOx loaded system. Main results of GOx-based delivery systems are summarized in Table 3.molecules-27-03836-t003_Table 3Table 3Summary of recent studies on GOx-based nanomaterials in cancer therapy.FormulationResultsStudy MethodRefs.Fabrication of GOx-poly(FBMA-co-OEGMA) nanogels (NGs)In a C8161 melanoma mouse model, NGs inhibited tumor growth 3.5-fold more effectively than the GOX (dose 100 mU) on 16 d post administration. NGs-treated mice exhibited 1.9-fold longer median survival times than GOX treated mice at the doses.IC_50_ of GOx made 24.4 ng/mL in C8161 cells.[96]Fabrication of Fe_3_O_4_@PPy@GOx nanocatalysts (NCs)The NCs (163.5 nm) exhibited cytotoxic activity in 4T1, HeLa, HUVEC cells and 4T1 tumor-bearing mice. The GOX activity was improved by photothermal-enhanced sequential Fenton nanocatalytic effect.70 mg/kg dose of Fe_3_O_4_@PPy@GOx resulted in efficiency in vivo.[100]Fabrication of GOx@ZIF@MPNadenosine triphosphate (ATP)-responsive NPsThe NPs (180 nm) exhibited antitumor activity in 4T1 cells and 4T1 tumor-bearing mice. The GOX activity was improved by ATP-responsive autocatalysis and acceleration of the Fenton nanocatalysis. 0.3 mg/kg GOx-contained NPs showed efficiency in 4T1 cells.[101]Fabrication of ZIF-8@GOx-AgNPs@MBNmultifunctional nanoreactor (Nr)The Nr (400 nm) showed high cytotoxic effect against HeLa cells and significant antitumor activity (96.8%, 200 μg/mL) in tumor-bearing mice. The Nr exhibited catalysis-enhanced synergistic starvation/metal ion poisoning cancer therapy.0.08 mg/mL of Nr showed 94% inhibitory effect in HeLa cells.[102]Fabrication of γ-PGA@GOx@Mn, Cu-CDs multifunctional NPsThe NPs (80 nm) exhibited in cytotoxic activity in 4T1 cells and tumor inhibition activity (~90–95%) in 4T1 tumor-bearing mice. The therapeutic action of GOX was improved by the PDT, PTT and checkpoint-blockade immunotherapy.100-150 μg/mL of NPs inhibited 4T1 cell by 82–91%.[103]Fabrication of PLL- and HA-modified GOx-loaded silica NPsMSNs-GOx/PLL/HA NPs reduced the initial tumor volume in mice bearing HepG2 cells. Modification with PLL/HA caused significant reduction in tumor volume.40 µg/mL dose reduced cell viability to <20%.[107]Fabrication of complex the GOX, PLL-g-PEG, and anti-PSMA antibodyModifying GOX with cationic copolymer and linking with anti-PSMA antibody efficiently inhibited PSMA-expressing prostate cancer cells.1 µg/mL dose reduced LNCaP cell viability to <40%.[108]

Hollow iron-tannic acid nanocapsules (HFe-TA), used for combined encapsulation of GOx and DOX (DOX/GOx@HFe-TA), have also been shown to exert starvation/chemodynamic synergistic effects [109]. Cell viability levels of 4T1 cells were reduced significantly after DOX/GOx@HFe-TA treatment, both in glucose-minus and RPMI media. For the first 14 days, the combined actions of GOx and DOX led to no statistical changes in tumor volumes in mice bearing 4T1 cells, whereas DOX@HFe-TA or GOx@HFe-TA treatments inhibited tumors by 35–40% compared to controls. PTX and GOx combination also resulted in synergistic effects when co-loaded in mesoporous silico NPs (MSNs-GOx) that were further modified with poly(L)lysine (PLL) and hyaluronic acid (HA) [107]. The treatment of tumor cells with this MSNs-GOx/PLL/HA nanodelivery system totally inhibited tumors. A lower efficiency level than that resulting from MSNs-GOx treatment could be attributed to the HA-based targeting. Additionally, MSNs-GOx/PLL/HA experiments resulted in pH-dependent drug release that could provide advantages inside tumor tissues. In contrast to this, PLL could facilitate endocytosis processes. It has been shown that the GOx cytotoxicity to prostate specific membrane antigen (PSMA)-expressing prostate cancer cells can be enhanced by modifying GOx with cationic co-polymer PLL-g-PEG, which improved the uptake levels in LNCaP cells [108]. 

Chemical conjugation of GOx with 4-formyl-N-(3-(2-(2-(3-methacrylamidopropoxy) ethoxy)ethoxy)propyl) benzamide (FBMA) and an oligo ethylene glycol monomethyl ether methacrylate (OEGMA) fabricated nanogel that was shown to enhance enzyme stability [99]. The developed nanogel with an average size of 802 nm efficiently inhibited tumor growth in a melanoma mouse model that equaled one-third of the volume of that resulting from free GOx. Moreover, the 60% survival rate was prolonged from 16 to 29 days, as compared to the survival rate resulting from free GOx treatment.

## 6. Asparaginase

Asparaginases are enzymes that catalyze the breakage of asparagine into aspartic acid and ammonia. By the action of this enzyme, the depletion of ammonia from glutamine also occurs [110]. For many years, these enzymes have been used as anti-neoplastic agents in cases of ALL [111]. Their main function is to deplete asparagine from the bloodstream by hydrolyzing it into aspartic acid and ammonia, resulting in tumor cells being retarded due to a lack of asparagine synthetase [112]. Moreover, lowered amounts of glutamine also enhance the dependence of tumor cells on asparagine for protein synthesis [113]. Apoptosis of ALL cells, induced by asparaginases, was shown to involve inositol 1,4,5-trisphosphate receptor signaling [114]. The triggering of intracellular Ca^+2^ by asparaginase further leads to the activation of caspase-3. Early medical practice evidence about asparaginases indicates that their use has side effects, such as the transient decrease in white blood cell counts and decreased spleen size [115].

Different approaches, such as obtaining protein fusions, conjugating with polymers, encapsulations in cell membranes or liposomes, forming dendrimers, and immobilizing with nanoparticles, have been used to protect and increase the anticancer efficacy of asparaginase [116,117]. Not all asparaginases result in cytotoxic effects in cancer cells due to reasons such as the optimum pH for some of these being around 7.5 [116]. Genetically modified forms of asparaginase, expressed in *E. coli* and other microorganisms [112,117,118], enabled the conservation of enzymatic activity at higher temperatures [119], lower glutaminase and conserved asparaginase activity [120], and enhanced resistance to degradation by trypsin [121].

Encapsulation of asparaginase fundamentally has been well studied but still requires more information to improve its efficacy by reducing its side effects. In medical practice, to treat ALL, its toxicity is usually managed by PEGylation, which can lead to hepatotoxicity, pancreatitis, hypertriglyceridaemia, thrombosis, or hypersensitivity [122]. Poly (lactic co-glycolic) acid has been used without cross-linking agents to encapsulate asparaginase to protect it during blood circulation. The effects of this nanosystem on blood and immune cells such as RBC, WBC, monocytes, neutrophils, and lymphocytes, were shown to be less severe compared to treatment with 5-fluorouracil. In vivo experiments showed no hemotoxicity and hepatotoxicity using this system. In an Ehrlich tumor model, the formulated nanosystem decreased tumor growth by ~50% compared to 5-fluorouracil. For 24 h, the cumulative drug release was as high as 80%.

An ATTEMPTS system trial to deliver asparaginase to tumor cells demonstrated increased cell permeability [123]. The TAT peptide, rich in arginine, was chemically conjugated to asparaginase via disulfide bonds and, unfortunately, increased its toxic effects. In mice bearing L5178Y lymphoma cells, the developed ATTEMPTS system prolonged the survival percentage by 2 days, resulting in an IC_20_ change from 15 days to 17 days (Table 4).

Liposomes are the most often-used encapsulation method utilized with this enzyme [112]. Early studies of liposome encapsulation using egg phosphatidylcholine, egg phosphatidylinositol, cholesterol, and other lipids demonstrated reduced toxicity and a prolonging effect of enzyme activity [124]. To this end, asparaginase-containing liposomes, formulated with lecithin and cholesterol, showed drug release percentages between 72% and 87% for 8 h, independently of the presence of positively or negatively charged molecules held in liposomes [125]. In addition, no significant differences in enzyme entrapment efficiencies were found in neutral or charged liposomes. Asparaginase encapsulation in lipids in chitosan enhanced this enzyme’s bioavailability and its anticancer effects [126]. The encapsulation of enzymes in lipid in chitosan nanosystems caused significant enhancement of the inhibition activity of H446 human lung carcinoma cells (~75%). The encapsulation was also established to increase the enzyme heat stability.

Nanogel formulation, achieved by chitosan-tripolyphosphate composition and stabilized by electrostatic interactions, demonstrated a high loading capacity (47.6%) and efficient encapsulation efficiency (76.2%) [127]. Encapsulation in chitosan-tripolyphosphate nanogel significantly enhanced the enzyme stability at high temperatures (60–70 °C) at pH 8.5. The in vitro half-life, linked with specific enzyme activity, in PBS (pH 7.4) and double-distilled water (pH 7.0) was prolonged from 30 to 72 h and from 24 to 15 days, respectively.

Asparaginase was found to incorporate into nanocomposites of chitosan and β-cyclodextrin, made of seven glucose subunits [128]. To this end, a mixture of chitosan and β-cyclodextrin was precipitated and sonicated before adding an asparaginase solution. The formed nanocomposites were separated by centrifugation and then freeze-dried. The anticancer efficacy of the formed nanocomposites was confirmed on PC3 prostate and U937 lymphoma cells, and concentration-dependent cell viability levels were observed. The enzyme-incorporated nanocomposites were found to have a higher impact on prostate cancer cells compared to lymphoma cells. β-cyclodextrin in tandem with gelatin was chosen as another option to immobilize asparaginase [129]. The morphology of this developed nanoformulation was shown to be spherical. At lower doses, the formulated nanocomposite demonstrated higher anticancer activity on HeLa cervical cancer cells relative to U87 glioblastoma cells. Lower efficacy on brain cancer cells was suggested to be due to tight junctions. β-cyclodextrin was also demonstrated to self-assemble hollow nanostructures with alginate-graft-poly(ethylene glycol) (Alg-g-PEG) and encapsulate asparaginase [130] (Table 4). Encapsulation revealed significantly higher relative enzyme activity at pH values of 7, 10, 11, and 12, but lowered activity at pH 8.5. In addition, the relative enzyme activity at 35 °C, 45 °C, and 55 °C significantly increased after encapsulation.

Cross-linker-free poly (lactic co-glycolic acid-based nanoformulation), loaded with asparaginase, was another efficient means that reduced the EAT tumor volume by 50% [131]. Nanoliposome, containing DSPE-PEG-200, was found to be efficient against several cancer cells. Loading enhanced toxic effect of asparaginase on LLC, HepG2, SK-LU-1, MCF-7, and NTERA-2, cells [132].

PEG113 and H-(2-hydroxypropyl) methacrylamide (HPMA) were also shown to be efficient composition mediums to load asparaginase [133]. The encapsulation in vesicles was carried in a water solution following centrifugation and re-suspension several times. The obtained nanovesicles were shown to be resistant to protease degradation and antibody binding (Figure 6). The main advantage of these developed nanovesicles was their highly reduced immunogenicity, but no significant changes in the cell viability of A549 human lung carcinoma cells were observed using these nanovesicles.

Genetically engineered *Salmonellae typhimurium* with controlled expression of asparaginase were directed to treat ALL, and this has demonstrated great potential [134]. This bacterium strain was engineered to express the enzyme within tumor cells via an inducible araBAD promoter (SL/pASN). The SL/pASN treatment was carried out on AsPC-1 adenocarcinomas, MC38 murine colon adenocarcinomas, and 4T1 breast cancer cells, and it showed at least 4-fold lower cell viability levels compared to treatments with PBS or ordinary bacterial strains, both of which were statistically irrelevant. The effects on Jurkat T-cell leukemia cell lines were even greater. Only slight differences in HEK 293 cell viability levels were observed between PBS and SL/pASN treatments. Animal experiments carried out on C57BL/6, BALB/c, and BALB/c athymic nu-/nu- mice bearing MC38, 4T1, and AsPC-1 cells, respectively, showed at least a 3-fold inhibition of tumor size for 30 days. In addition, the survival rate was greatly prolonged, or no death was observed following treatment with SL/pASN.

Biomimetic camouflage strategies have attracted increased attention over the last decade in cancer therapy [135]. Among these DDSs, the RBC membrane is of unique interest because of higher resources relative to other locations. Loading asparaginase on RBCs has been well studied and has even reached manufacture stages [136]. Loading *E. coli* asparaginase in homologous RBCs was carried out in an automated process consistent with personalized medicine principles. The drug regimens were ordered based on patient weight and blood type, as well as irregular antibody screening in patients. Delivery and conformance rates were found to be 100%, and side effects in patients were significantly reduced, relative to care treatment standards.

One of the approaches to increase the bioavailability of asparaginase includes its immobilization on Au NPs [137]. In this manner, conjugates of Au NPs and SH-PEG-COOH were obtained (GNPs-PEG-COOH) in water and further linked with CPP-asparaginase fusions coupled with FITC (GNPs-PEG-RGD-asparaginase). Cell viability mean values in MCF-7 breast cancer cells using the developed GNPs-PEG-RGD-asparaginase system were reduced by a third, which were significantly lower than that resulting from the treatment with free asparaginase and Au NPs. Additionally, the cell permeability of the developed system significantly enhanced, and the mitochondrial membrane potential significantly lowered compared to those resulting from free asparaginase and Au NPs, which were not significantly different from the controls. Moreover, cytochrome C release was significantly increased by this conjugation. Main results of ASP-nanomaterials are summarized in Table 4.molecules-27-03836-t004_Table 4Table 4Summary of recent studies of ASP-based nanomaterials in cancer therapy.FormulationResultsSize (nm)EE and LC LevelIC 50 ValueRefs.PTD-modified ATTEMPTS system for ASPThe inclusion of TAT peptide was not able to enhance the cytotoxic effect of ASP in vivo.

0.0100 UI/mL (L5178Y)[123]Encapsulation in positive liposomeEnhanced level of physical stability and in vitro cytotoxicity.35.2 ± 4.52.39% (LC)50 µg/mL (EAC) [125]Formulation of chitosan modified lipid nanoparticlesSignificantly (14%) lowered IC_50_ value on H446 lung carcinoma cells.426.60 ±36.3466.47 ± 2.81 (EE)
[126]Encapsulation in chitosan/TPP nanosystemsASP with high LC became more resistant to high temperature and alkaline condition.340 ± 1276.2% (EE)47.6% (LC)
[127]β-cyclodextrin-chitosan-ASP nanobiocompositeObtained nanobiocomposite revealed a four-fold activity on PC3 over U937 cells.40–80
125 µg/mL (PC3)[128]β-cyclodextrin-gelatin-ASP nanobiocompositeDeveloped nanobiocomposite showed better activity on Hela than U87 cells.59–81.6av. 74.1 
62.5 µg/mL (Hela)[129]Hollow NPs of Alg-g-PEG and cyclodextrin with ASPEncapsulation significantly increased the enzyme stability in an acidic condition.av. 46737–80% (EE)
[130]Poly (lactic-co-glycolic) acid nanoformulationEncapsulation in PLGA significantly inhibited EAT-tumor in mice.195 ± 0.280.23 (EE)10% (LC)
[131]Encapsulation in liposome containing DSPE-PEG-200Encapsulation in liposomes significantly reduced LLC-tumor volume in mice.93.03 ± 0.4953.99 (EE)0.23 UI/mL (LLC)[132]Immobilization of ASP-RGD on to Au NPs via PEGRGD peptide-targeting enhanced anticancer efficacy on MCF-7 cancer cells.20–50av. 29.24
89.8 µg/mL (MCF-7) [137]Immobilization of ASP on to Au NPs Obtained nanobiocomposite showed high toxicity to A549 and A2780 cancer cells.20–50 
62.5 µg/mL (A549) [138]Cerium-selenium nanobio-composite with ASPCerium-selenium nanobiocomposite caused synergistic effect with ASP.60–90
125 µg/mL (A549)[139]Immobilization to magnetic NPs of SiO_2_, Fe_3_O_4_, PVDMALonger polymer chain was concluded to be more favourable for enzymatic reaction.app. 20–3032% (LC)
[140]

In another trial, direct immobilization of FITC-labeled fungal asparaginase on Au NPs was carried out using glutaraldehyde [138]. The anticancer efficacy of the obtained nanobiocomposites was assessed in A2780 ovarian and A549 lung cancer cells. When up to 100 µg/mL of this nanocomposite was used, the cell viability of ovarian cancer cells was reduced to only ~80%. However, this nanocomposite reduced the cell viability levels of lung cancer cells by <45% at 125 µg/mL. The developed novel DDS thus could serve as a good candidate for combating lung cancer.

Cerium-selenium nanocomposites have been demonstrated to bind asparaginase [139]. The obtained nanobiocomposite was shown to have anticancer efficacy in A549 lung cancer cells in a concentration-dependent manner. Cellular internalization of asparaginase-bound nanocomposites labeled with FITC was observed by fluorescent microscopic studies. Poly(2-vinyl-4,4-dimethyl azlactone) (PVDMA)-modified iron NPs (via tetraethyl orthosilicate) could immobilize asparaginase onto their surface and have been developed as a magnetic carrier [140] (Table 4). Longer chains were found to support enzymatic activity. The fabricated nanoparticles were claimed to be efficiently used to immobilize this enzyme. For 10 weeks, over 70% of the enzyme activity was retained, and these NPs were found to have long-term stability.

## 7. Exotoxin

Toxins are defined to be poisonous proteins, as well as secondary metabolites that are resistant to antibiotics or chemical agents. These compounds are produced by both prokaryotes and eukaryotes [141]. Microbial toxins are expected to be easy to work with since these organisms and toxins can be genetically modified. Among microbial toxins, the ones of bacterial origin have been more completely investigated, especially for efficacy against cancer [142]. Exotoxin A, isolated from *Pseudomonas aeruginosa*, has been found to be of high interest as a potential anticancer agent. This toxin inhibits ADP-ribosyltransferase activity and thus contributes to the blockage of protein synthesis [143]. A 38-kD protein fragment, identified in *P. aeruginosa* (PE 38), was demonstrated to have high activity against cancer cells [144] and has served as a useful platform for further development. A recombinant form of PE 38, Anti-Tac(Fv)-PE38, was demonstrated to be an efficient therapeutical tool against CD25-overexpressing B cell malignancy [145]. Early analysis identified three domains in the Pseudomonas exotoxin A (PE), namely, recognizing receptors, translocating the cytosol, and ribosylating the elongation factor [146]. The catalytic domains of exotoxin, termed PEIII or PE3, inhibit ADP-ribosylation and thus damage protein synthesis. Extensions in catalytic subunits caused its inactivation, but no effects were determined in protein synthesis [147]. The significance of exotoxin or its derivatives was concluded to be that they deplete oncogenic signaling molecules and cancer cell-secreted growth factors [148].

PE was found to act as a vehicle that can stimulate the protective response of cytotoxic T-lymphocytes to antigens in vivo [149]. To demonstrate this concept, an ovalbumin protein fragment was inserted into a 64-kD nontoxic mutant PE. Thus, nontoxic PE was fused to ovalbumin that was used to vaccinate mice. This protein fusion generated CD8+ lymphocytes that lyse ovalbumin epitope-expressing murine cells. In another work, the catalytic domain of PE (PEIII) was loaded in chitosan microparticles (1.09–1.46 µm), cross-linked by tripolyphosphates [150] (Table 5). The release of the encapsulated 28.7-kD protein PEIII from the developed microparticles was correlated with the concentration of the cross-linking agent.

Experiments carried out on *Drosophila melanogaster* S2 cells determined their sensitiveness to PE at picomolar concentrations [151]. Earlier studies that established an increase in caspase activity further clarified the dependence of the toxin-mediated death of S2 cells on diphtamide, modification of elongation factor, and terminal caspase in insects.

A PE conjugate with folate has been used to deliver this toxin to folate receptor-overexpressing cells [152]. This exotoxin-folate conjugate showed a 10-fold higher potency against folate-deficient HeLa cells, compared to a momordin-folate tandem toxin. Additionally, FDKB tumor cells have been demonstrated to have high uptake levels of toxin-folate conjugates. The increased toxicity of this conjugate was linked to the translocation domain of this toxin.

Another strategy to deliver PE in cancer cells utilized the targeting of granulocyte-macrophage colony-stimulating factor receptor (GM-CSFR), which is overexpressed in some types of leukemia and solid tumors [153]. To this end, PE was genetically fused with a peptide sequence possessing binding affinity to GM-CSFR. Recombinant PE exotoxin showed very high potency against LS174T and SW403 colon cancer cells and N87 and HTB103 gastric cancer cells. The IC_50_ values of this toxin were 2.2, 0.9, 0.45, and 9.5 ng/mL, respectively in LS174T, SW403, N87, and HTB103 cells. Recombinant forms of DT were found to have greater potency on promyelocytic (HL60), erythroleukemia, and monocytic (U937) cells, with IC_50_ values 0.4, 0.02, and 0.04 ng/mL, respectively. There were no significant differences in the expression levels of GM-CSFR in the studied cells that were observed in the experiments that were used to determine these IC_50_ values. Thus, different toxins were found to have different potencies on solid and hematopoietic tumor cells independent of the receptor expression level.

Linking anticancer drugs to antibodies is one of the most efficient approaches in drug delivery. In this discipline, a few studies have been carried out on PE. In a trial, PE and DT were conjugated to trastuzumab, a type of monoclonal antibody specific for breast and stomach cancers [154]. This conjugation reduced the viability levels of SK-BR-3 breast cancer cells by 3-fold compared to PE. Similar results were observed with DT. Obtaining conjugates of toxins has been proposed as an efficient approach to reduce the therapeutic dose required of an antibody. Noncovalently linked combination of human or mouse antibodies with an immunoconjugate of a truncated PE with alpha crystallizable fragment (α-Fc) antibodies (α-Fc-ETA′) could inhibit the growth of antigen-expressing tumor cells [155]. Cytotoxic effects were highly dependent on antibody conjugates, α-Fc-ETA′, at low concentrations.

Another antibody-related work developed a conjugate of a truncated PE isoform with human κ-light chain binding antibody fragments [156]. The obtained conjugate could specifically inhibit the proliferation of antigen-expressing cells and triggered the apoptosis of target cells. Thus, covalently linked antibodies were turned into immunoconjugates with CD22-targeting ability and provided exotoxin with enhanced efficacy [157]. Both in vitro and in vivo experiments carried out on B cell malignancies pointed toward improved toxic effects.

The fusion of anti-mesothelin Fv (SS1) with PE38 [SS1(dsFv)PE38] has been efficiently used to target mesothelin, a 40-kD tumor differentiation antigen [158] (Table 5). Minor effects of this toxin were detected in patients with ovarian cancer and mesothelioma. More than half of the patients included in this study had a stable disease. The maximum tolerated dose of this toxin was 45 µg/kg when administered intravenously. Pleuritis was shown to be a dose-limiting side effect. Further development in this discipline resulted in the fusion of SS1 Fab fragment with truncated 24-kD exotoxin (PE24) [159]. This new fusion, termed RG7787, could inhibit lung cancer cell lines at picomolar doses. Side effects such as hepatotoxicity and vascular leak syndrome in rodents were improved 10-fold compared to SS1P alone. The tumor growth in mice bearing NCI-H596 xenografts was totally inhibited by RG7787 at a dose of 2 mg/kg. A three-cycle administration led to a several-fold reduction in tumor volume in tumors with an initial volume of 500 mm^3^.

A recombinant immunotoxin, termed RFB4(dsFv)-PE38 (BL22), reached the second phase of clinical trials for hairy cell leukemia (HCL) by 2008 [160]. The potential of this immunotoxin, consisting of an anti-CD22 antibody and a 38-kD Pseudomonas exotoxin, was demonstrated, and BL22 has been recommended for further study for efficacy against chemo-resistant HCL. Other immunotoxins that reached clinical stages have been reviewed by Wolf and Elsässer-Beile [13]. These authors demonstrated a structural map, and functional domains were determined in the exotoxin A: ADP-ribosyltransferase catalytic, receptor-binding, and translocation domains (Figure 7).

CD22-targeting is a widely used strategy that has provided exotoxin with enhanced efficacy [161]. Both in vitro and in vivo experiments carried out on B cell malignancies pointed toward improved toxic effect. Kreitman et al. utilized CD22 targeting to combat HCL [162], lymphoma, and leukemia [163], while earlier authors used an anti-CD25 strategy against lymphoma and leukemia [164]. PE-based immunotoxins, developed by Pai et al., have even reached clinical phase I for use against ovarian cancer and adenocarcinoma by targeting ovary [165] and LeY carbohydrate antigens [166], respectively. Later, the efficiency of this LeY antigen-targeting strategy was demonstrated by another group of researchers [167]. Additionally, colon, pancreas, and head/neck cancer cells were demonstrated to have 80–100% of the antigen expression of these cells in more than 60% of patients.

The fusion of the interleukin-4 (IL-4) sequence with truncated PE38 represents another immunotoxin that has reached clinical stages. It was used against IL-4-expressing glioblastoma [168] as an intratumoral infusion and other cancer cells expressing IL-4 receptor. In one work, the effect of IL-4-fused PE NBI-3001 exotoxin was studied in 25 patients diagnosed with glioblastoma multiforme [169]. This toxin was found acceptable as administered intratumorally to patients with malignant glioma. In another study, NBI-3001 was administered intravenously to patients with renal cell carcinoma (RCC) and NSCLC [170]. The effects of this toxin on RCC were rather negative. Unfortunately, stable disease continued in 67% of patients. Moreover, drug administration enhanced the hepatotoxicity. Main results of PE-based toxins in cancer therapy are summarized in Table 5.molecules-27-03836-t005_Table 5Table 5Summary of studies of PE-based toxins in cancer therapy.FormulationResultsDose ActionsRefs.Effect of PE exotoxin fused with ovalbumin on mice bearing EG7 cellsTumor growth was greatly inhibited in ovalbumin expressing cells in vivo. No changes were determined in the tumor volume in non-ovalbumin expressing cells in vivo.100 µg dose led to ~20% lysis of EG7 cells.[149]Loading of PEIII in chitosan microparticles cross-linked TPP with ~1.09 µm sizeExtension of cross-linking time reduced the drug release. The drug release was found to enhance with the increase in sonication power > 45 W.60–80% toxin was found released for the first few hours.[150]Phase I study of anti-mesothelin dsFv-PE38 (SS1P) in mesothelin-expressing cancersDose-limiting toxicity was linked normal pleural mesothelial cells that express mesothelin. No pericardial toxicity was observed despite mesothelin expression on normal pericardial cells.Maximum tolerating dose made 45 µg/kg every other day[158]Fusion of PE24 with humanized SS1 Fab fragment (RG7787)A three-cycle treatment with RG7787 led a reduction in initial 500 mm^3^ tumor volume by more than half for 110 days in NCI-H596 tumor model.2–3 mg/kg dose of RG7787 was more efficacy than SS1.[159]Phase II trial of recombinant RFB4(dsFv)-PE38 (BL22) in chemoresistant HCLSingle cycle of BL22 was found highly effective and without serious toxicity. Selective retreatment enhanced complete remission rate to 64% with no dose-limiting requirement.40 µg/kg dose (every other day) caused 25% complete remission rate[160]Phase I trial of PE 40 (BR96 sFv-PE40) with advanced solid tumor in patientsRapid drug clearance made 11 days in many patients in association with Human Antitoxin Antibody. Partial tumor responses were observed for eight weeks.0.641 mg/m^2^ with gastrointestinal dose-limiting toxicity.[167]Phase I trial of Il-4-fused PE (NBI-3001) in tumors expressing IL-4 receptorHepatotoxicity was the main side-effect that prevented escalating dose of NBE 3001. Fatigue, headache, arthralgia were reported as main adverse events. The toxin was detected in less than 5% of patients.8–16 mg/m^2^ daily dose 5 times every 28 days caused no dose-limiting toxicity[170]Phase I trial of recombinant erb-38, containing Fv portion of e23 monoclonal antibodyObserved hepatotoxic effects were suggested linked with the expression of erbB2 on hepatocytes. Targeting with erbB2 was concluded to likely cause side effects in liver.1.0 and 2.0 µg/kg doses caused hepatotoxicity in patients.[171]Phase study of ScFv(FRP5)- ETA against erbB2-overexpressing tumor cellsLocal therapy of ScFv(FRP5)- ETA was concluded to be effective against erbB2-expressing tumor. Retreatment was suggested to be effective as antibodies recover in patients.0.6–6mg dose caused shrink in 60% cases of the study.[172]Phase I study of scFv(FRP5)-ETA in erbB2-overexpressing metastatic cancersResults indicated maximum tolerated dose 12.5 µg/kg can be administered to erbB2-overexpressing tumors. The main side-effect was hepatotoxicity due to erbB2 expression in hepatocytes.10 µg/kg dose was suggested no to induce side–effects.[173]

The overexpression of ErbB2 in cancer cells was utilized to develop a PE-fused immunotoxin with the Fv fragment portion of the e23 monoclonal antibody [171]. Results from phase I studies, carried out in patients with breast cancer, showed that targeting anticancer agents via antibodies to ErbB2 might cause toxicities due to its expression in normal cells. Hepatotoxicity was detected in all patients included in this study. Targeting ErbB2-overexpressing cells was performed in nine patients with breast cancer at a site of metastasis in the skin, lungs, lymph nodes, liver, and chest wall and in two patients with melanoma colorectal cancers [172]. At 8–11 days post intratumoral administration of scFv(FRP5)-ETA toxin into cutaneous lesions, among the three patients, two patients developed antibody responses and one achieved complete remission of tumor nodules. Furthermore, a dose-limiting toxicity study of this recombinant toxin was carried out in 18 patients with ErbB2-overexpressing metastatic cancers [173] (Table 5). Thus, a maximum tolerating dose was established at 12.5 µg/kg as administered intravenously. Hepatotoxicity was found as a dose-limiting side effect, as well.

EGFR binding ligand is another useful ligand that has been used in fusion to PE 38 exotoxin [174]. This study was carried out in 20 patients, 16 of which were afflicted by glioblastoma multiforme. Recurrent disease was observed in almost all patients when the drug was administered at a dose of 1–4 µg. Signs of non-specific toxicity were, however, not detected. In earlier trials, the ability to bind EGFR did not lead to antitumor activity in patients with resistant bladder cancer at doses of up to 9.6 µg/week for 6 weeks [175]. However, clinical improvements were observed in patients with carcinoma. In another study, PE was fused with both heparin-binding domain (HB) and EGF receptors or with HB only [176]. Both domains were shown to be useful for the efficiency of this toxin, despite the main role being attributable to EGF receptors.

The most recent review paper on *P. aeruginosa* exotoxin discussed the PE-based immunotoxins that have reached the second and third phases of clinical trials [177]. Some of these drugs were combined with lower molecular anticancer agents. For example, a recombinant exotoxin, termed LMB-100, has been combined with nanoalbumin-bound PTX to target pancreatic adenocarcinoma [178]. The maximum tolerated dose of this anticancer agent was shown to be 65 µg/kg. In another work, recombinant immunotoxin administration was combined with pentostatin and cyclophosphamide to improve its targeting abilities [179]. Thus, these two chemicals prevented the immunotoxicity of this immunotoxin in mice.

The fusion of PE with scFv, which is an anti-epithelial cell adhesion molecule, has been developed by Viventia Biotech Inc. and has passed phase II, III clinical trials in patients with head and neck squamous cell carcinoma and bladder cell carcinoma [180].

## 8. Diphtheria Toxin

DT is a 58-kD protein secreted by *Corynebacterium diphtheriae* [181]. In the human body, it causes the inhibition of protein synthesis by transferring ADP-ribose groups from NAD to elongation factor 2 (EF-2). Three domains have been demonstrated in a crystal structure of this protein: a catalytic domain, a transmembrane domain, and a receptor binding domain [182]. The transmembrane domain consists of mainly nonpolar amino acids and has been found to participate in pH-triggered translocation. Based on an analysis of the crystal structure of DT without nucleotide, an important role of these hydrogen bonds in the binding of its substrate (the elongation factor) to the catalytic domain was suggested [183]. Studies on crystals of both monomeric and dimeric DT revealed that receptor binding requires the cleavage of disulfide bonds between the transmembrane and catalytic domains [184]. A low pH value is considered preferable for the formation of an open monomer structure, which eases the penetration of the transmembrane domain.

Improving strategies for the use of DT includes both obtaining its conjugates with antibodies and fusing it with other peptides or fragments of proteins [185]. One of the more efficient trials toward improving DT obtained its fusion with IL-2 [186]. This toxin conjugate could suppress the growth of hepatocellular carcinoma cells previously treated with retinoic acid, which was used to induce hypoxia and IL-2 receptor expression (Table 6). The efficiency of a recombinant fusion of DT with IL-2, termed denileukin diftitox (DD), against anaplastic large cell lymphoma, was demonstrated in a recent case report [187]. A long-term treatment revealed that DD might cause some negative effects, such as capillary leak syndrome and infusion reactions, but proper pre-infusion (systemic steroids) and post-infusion regimens minimized its toxicity. Earlier, DD was found to not cause regression of metastasis in patients with metastatic melanoma [188]. One of the more recent developments with DD was the substitution of a single amino acid (V6A), which was shown to reduce the lethal dose in mice by 3.7-fold [189]. A sequential approach significantly improved the inhibition of tumor growth in melanoma-bearing mice. Further clinical studies are needed to clarify the efficiency of sequential substitution using this toxin.

In a number of cases, DT has been expressed in targeted cells by delivering genes using different approaches. For example, Adeno-associated virus vector expressing DT A chain was found to efficiently inhibit pancreatic carcinoma cells in vitro [190]. Toxic effects were not observed in normal cells. In vivo experiments, carried out in a xenograft mouse model bearing pancreatic cancer cells, demonstrated growth inhibition in a dose-dependent manner. In another work, the DT A gene was delivered to prostate cancer cells using a similar strategy [191]. The gene expression in BCL-2 apoptosis regulator-expressing androgen-independent prostate cancer cells was linked to halved tumor volume in mice bearing PC-3 cells. Chorionic gonadotropin promoters have been used as another delivery agent for the DT A chain gene [192]. The efficiency of gene delivery and its expression were also assessed using the inhibition of luciferase protein synthesis in ovarian cancer cells in vitro. Additionally, retroviral packaging was also used to deliver DT-A chain [193]. It was demonstrated that retroviral packaging cells, derived from TK-NIH/3T3 cells, were partially resistant to DT, and production of an attenuated version of DT-A gene could be efficiently used for suicide gene therapy. Another approach to deliver the DT A gene (pG-DTA) to cancer cells utilized loading it in vitamin E succinate-grafted PEI cores with an RGD-modified PEGylated hyaluronic acid shell [194]. In vitro analysis carried out on B16-F10 melanoma cells demonstrated greatly improved efficiency of this gene after encapsulation in micelles. The administration of gene-loaded nanomicelles reduced the tumor volume in mice bearing B16-F10 cells by 4-fold compared to those treated with free pG-DTA. Moreover, no regressions were observed in the survival rates over the next 22 days after encapsulated gene administration. The expression of the DT gene controlled by H19 regulatory sequences was chosen as another possible approach to combat ovarian cancer [195] (Table 6). Controlled by H19, the RNA present in ovarian cancer cells at high dose and almost undetectable in adjacent normal cells resulted in a 40% inhibition of the growth of ectopically developed tumors. This approach could be considered efficient against ovarian cancer, as 90% of patients with ovarian cancer ascites fluid present with H19 RNA. Further development led to a DNA-based therapy using DT A with H19 gene regulatory sequences for pancreatic cancer [196]. The efficiency of H19 regulation over and with DT A was also studied by comparing it with H19-regulated luciferase expression (Luc-h19) in a heterotropic model in mice. The tumor volume in mice bearing CRL-1469 pancreatic carcinoma cells was inhibited more efficiently by over 3-fold compared to controls following the administration of DT A-H19. Animals with metastasis following the administration of DT A-H19 resulted in 3 out of 9 being alive on day 11, whereas multiple abdominal metastases were observed in animals treated with Luc-H19. These results imply the high efficiency of utilizing H19 regulation with DT A. The efficiency of BC-819 (DT A-H19) plasmid was earlier established against different lung cancer cell lines, including A549, NCI-H358, and NCI-H460 cells with a >90% reduction in cell growth [197]. Its effects in mice with xenografts and lung metastasis models demonstrated that aerosolized PEI/BC-819 could significantly reduce tumor growth, while it had no effects on tumor metastasis.

The delivery of DT A genes has reached clinical stages in a few cases. DNA plasmid for a gene regulated by the H19 promoter sequence (BC-819) has made it to phase II clinical trials for use in cases with invasive bladder tumor [198]. The results implied the relatively good efficiency of this plasmid. A third of the patients in this study had complete tumor ablation, and others had no new tumors over the next 3 months.

The next stage of the development of DT-based prodrug forms involved fusing it with targeting sequences. In one work, T22-DITOX-H6 and T22-PE24-H6 sequences were developed by Geneart (Thermofisher), and recombinant proteins were obtained on this basis to target CXCR4-expressing cells [199]. T22 is a translocation domain, DITOX and PE24 are the sequences of DT and PE including their catalytic and translocation domains, and H6 is a hexa-histidine domain (Figure 8). Main results of DT-based toxins in cancer therapy are summarized in Table 6.molecules-27-03836-t006_Table 6Table 6Summary of recent studies of DT-based toxins in cancer therapy.FormulationResultsDose ActionsRefs.DT fusion with IL-2 (DAB_389_-IL2) that target IL-2-overexpressing hypoxic HCC cellsThe combination DAB_389_-IL-2 and retinoic acid caused the suppression in hypoxic HCC compared to treatments with either DITOX-IL-2 or retinoic acid.5 µM (SNU-475) 20 µM (HepG2) caused significant changes[186]Construction of V6A derived from DAB-IL-2 with single amino acid substitution (sDAB-IL-2)Combined action of sDAB-IL-2 with anti-PD-1 antibodies inhibited tumor growth ~5-fold more efficiently compared to sDAB-IL-2 treatment in B16F10 melanoma model.5 µg dose on day 7 and 10 significantly reduced B16F10 tumor volume[189]Plasmid of DT with H19 regulatory sequences that target ovarian cancer ascites fluidDTA-H19 plasmid more significantly inhibited tumor volume in mice bearing ES-2 cells compared to luciferase-H19 plasmid complexed with PEI.25 µg dose was administered four times with two-day intervals[195]Construction of DITOX-H6 fusion with T22 that target CXCR4-overexpressing cellsT-22-DITOX-H6 reduced tumor size of CXCR4 expressing HeLa cells ~6-fold compared to control. T22 showed efficacy wih DITOX-H6 and PE-24 toxins.By 10 µg (3 times a week) 8 doses in tumor model[199]Fusion of DITOX-H6 with T22 that target CXCR-4-overexpressing cells (T22-DITOX-H6)Self-assembling T22-DITOX-H6 NPs efficiently targeted AML cell lines that overexpress CXCR-4 and thus revealed antineoplastic effect.10 µg dose (10 times) potently blocked AML in bone marrow[200]Construction of DT with VEGFThat target vascular endothelial growth factorThe treatment with DT-VEGF significantly inhibited the tumor volume in mice bearing HPAF-2 (four-fold) and AsPC-1 (two-fold) in mice.100 µg/kg dose (every other day) after 3 days of tumor implantation[201]

The cell viability levels of Panc-1 human pancreatic cancer cells and HeLa cervical cancer cells treated with T22-DITOX-H6 and T22-PE24-H6 recombinant fusions reduced by more than half compared to controls, while exotoxin fusions showed higher efficiency. The administration of T22-DITOX-H6 inhibited tumor growth in mice bearing colorectal cancer cells by more than 5-fold compared to buffer-treated controls. T22-PE24-H6 administration led to a 2.3-fold reduction in colorectal cancer cells. Furthermore, T22-DITOX-H6 has been shown to induce apoptosis in CXCR4-expressing leukemia cells [200]. In an acute myeloid leukemia model in mice, T22-DITOX-H6 could block the spread of AML cells. No toxicity was detected in healthy cells. This recombinant toxin was suggested to be a promising drug tool against CXCR4-expressing leukemic cells.

The fusion of DT with VEGF was created to target the vasculature associated with pancreatic cancer [201] (Table 6). At high drug concentrations, in excess of 1000 ng/mL, the tumor volume of HPAF-2 and AsPC-1 pancreatic cancer cell models was reduced by 76% and 53%, respectively. However, a low concentration (10 ng/mL) could reduce the growth of HUVEC in vitro. Additionally, this recombinant toxin reduced the tumor spread of HPAF-2 (9-fold) and AsPC-1 (2-fold) cells very efficiently. Heparin-binding EGF-like growth factor (HB-EGF) was found to bind to DT by introducing changes to its receptor binding domain. [202] It was concluded that this developed toxin could be efficiently used against HB-EGF-overexpressing tumor cells.

## 9. Conclusions

Protein toxins have been shown to be efficient anticancer drug tools. Ideally, a small number of molecules have been shown to be enough to cause the death of cancer cells. This feature provides protein toxins advantages over lower molecular weight drugs. Their application, however, is hampered by a number of obstacles such as being attacked by immune cells, their heavy weights and bulky volumes, aggregation during storage, and low productivity. Therefore, different approaches have been used to improve their efficiencies, including fusion with peptides that provide targeting and penetrating functions, encapsulation in liposomes and nanoparticles, different releasing options that activate at lower pH, linking (covalently and noncovalently) with transport proteins such as albumin or lactoferrin, providing their expression in targeted cells by delivering appropriate genes, and utilizing physical-stimuli responsive systems. These different modifications have resulted in some of these toxins reaching the market. Ongoing studies should focus on the application of these protein toxins with other drug forms that might provide synergistic effects.

## Figures and Tables

**Figure 1 molecules-27-03836-f001:**
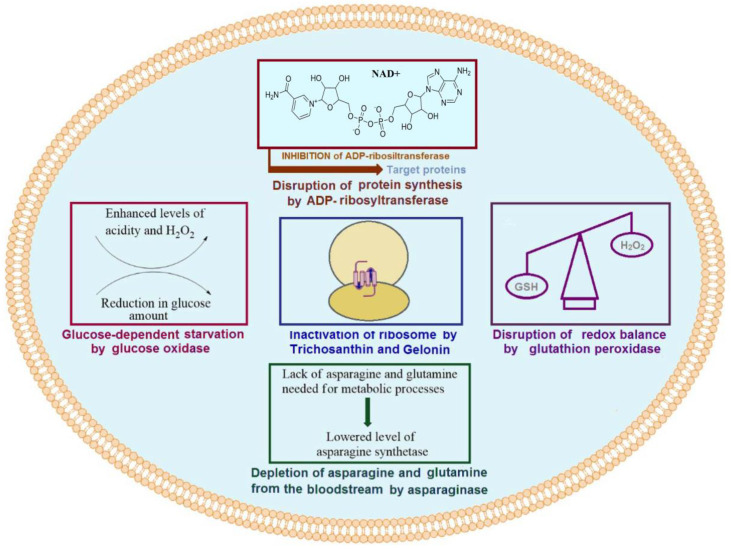
Mechanisms of action of selected enzyme drugs used in cancer therapy.

**Figure 2 molecules-27-03836-f002:**
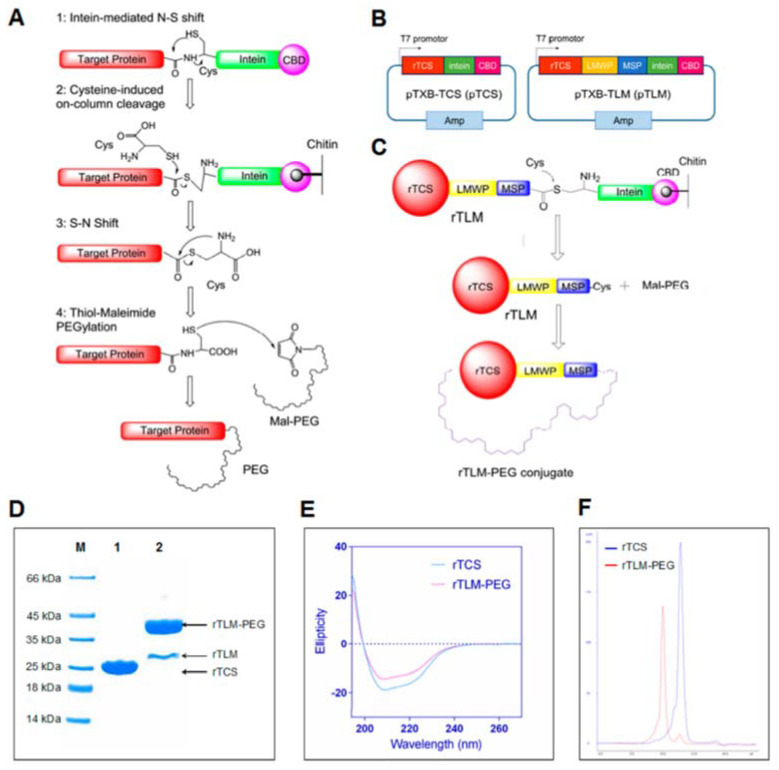
Obtaining PEGylated conjugate of TCS using an intein-mediated pathway. Reprinted with permission from ref. [42]. Copyright 2022 American Chemical Society. (**A**) Obtaining a TCS-PEG conjugate. (**B**) Recombinant plasmid construction. (**C**) Obtaining TCS-LMWP-MSP-PEG conjugates. (**D**) PAAG electrophoresis of conjugates. (**E**) Circular dichroism results. (**F**) Size exclusion chromatography results of these conjugates.

**Figure 3 molecules-27-03836-f003:**
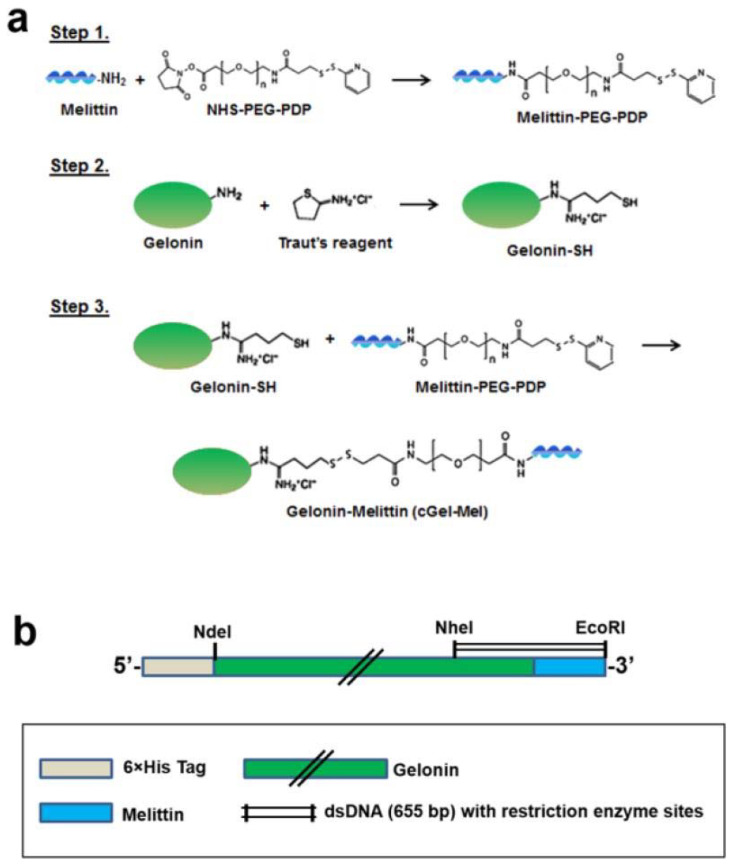
Synthesis of fusion of Gel with melittin. Reprinted with permission from ref. [58]. Copyright 2022 Springer Nature. (**a**) Chemical conjugation of Gel and melittin. (**b**) Schematic representation of Gel-melittin fusion gene contained in pET28a vector.

**Figure 4 molecules-27-03836-f004:**
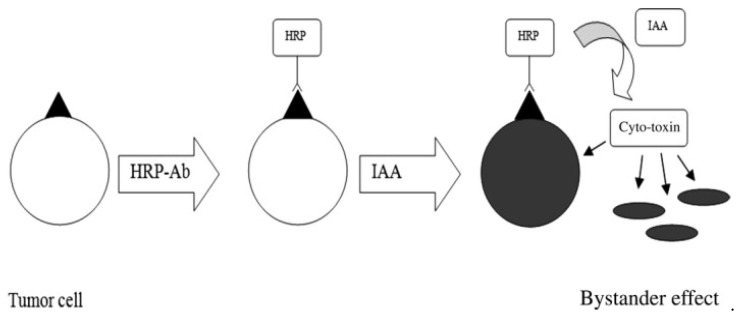
Utilization of HRP in antibody-directed enzyme prodrug therapy with IAA. Reprinted with permission from ref. [85].Copyright 2022 Wiley.

**Figure 5 molecules-27-03836-f005:**
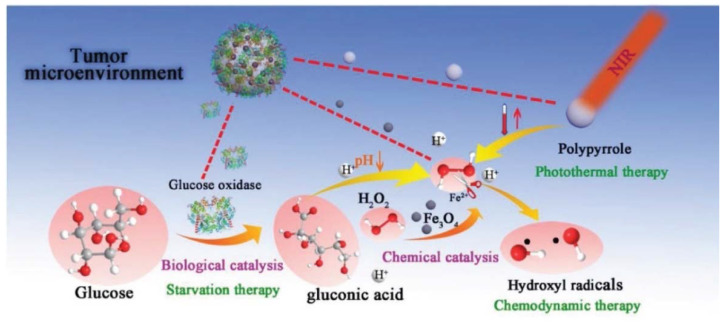
Anticancer effects exerted by Fe_3_O_4_@PPY@GOx nanocomposites. Reprinted with permission from ref. [100]. Copyright 2022 Wiley.

**Figure 6 molecules-27-03836-f006:**
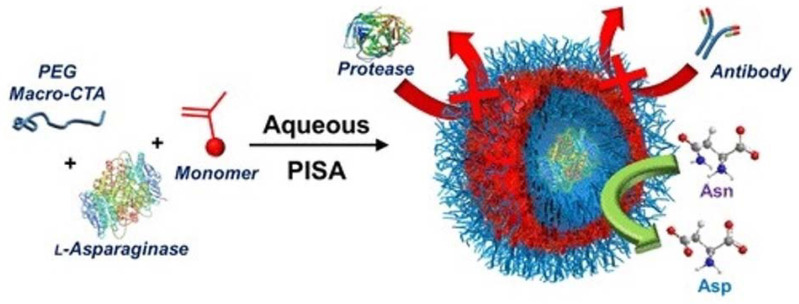
Asparaginase loading in nanovesicles by aqueous polymerization-induced self-assembly (PISA). Reprinted with permission from ref. [133]. Copyright 2022 American Chemical Society.

**Figure 7 molecules-27-03836-f007:**
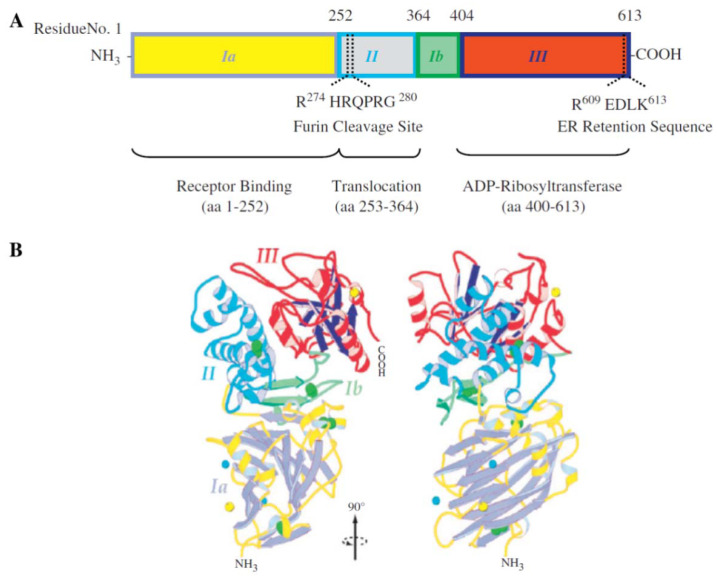
Structural map and functional domains of Pseudomonas exotoxin A. Reprinted with permission from ref. [13]. Copyright 2022 Elsevier. Different domains are shown in different colors both in (**A**) the structural map and (**B**) the protein ribbon structure.

**Figure 8 molecules-27-03836-f008:**
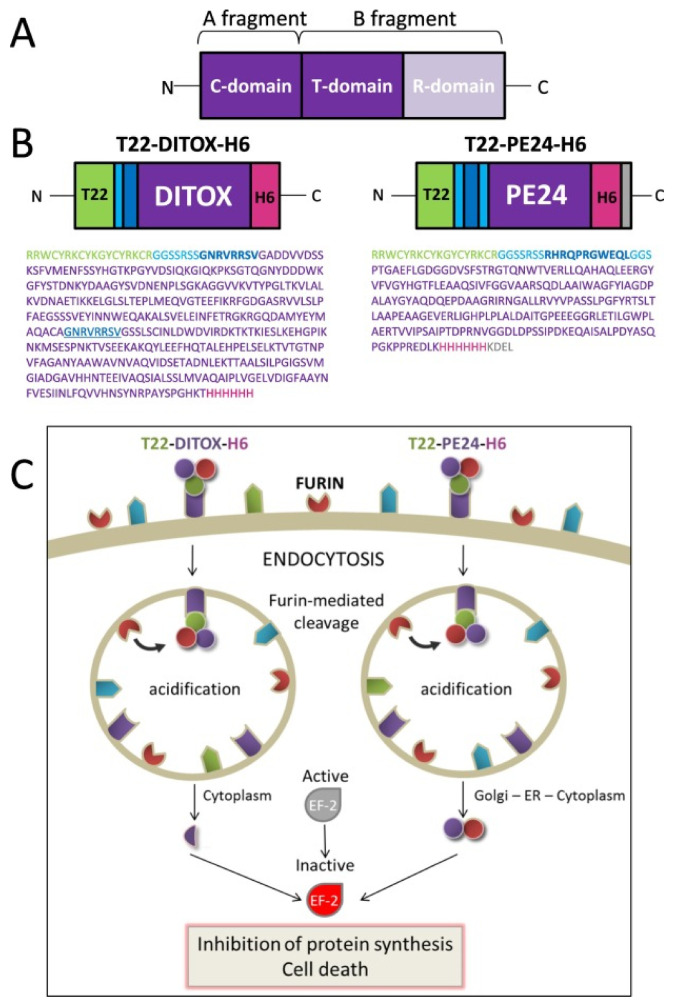
Development of recombinant toxin fusions to improve the efficiencies of DT and PE. Reprinted with permission from ref. [199]. Copyright 2022 Elsevier. (**A**) Structural map of native DT and PE. Catalytic, translocation, and receptor-binding domains are depicted by C-, T-, and R-, respectively. (**B**) Structural maps of these recombinant toxins developed based on DT and PE, and their amino acid sequences. With light- and dark-blue colors, linker regions and furin-cleavage sites and their sequences demonstrated, respectively. Green residues represent the translocation domain. (**C**) Possible mechanisms of actions of recombinant toxins in CXCR4-expressing target cells. Furin-mediated release of catalytic domains eases biodistribution and penetration into cells.

## Data Availability

Data is contained within the review article.

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
