# Peer review of "Advances on Delivery of Cytotoxic Enzymes as Anticancer Agents"

_molecules, 2022, doi:10.3390/molecules27123836_

Round 1

Reviewer 1 Report

What is “T Enzymes” (line 35 page1)? Is it typo?

Most of the cited works are devoted to obtaining a protein conjugate in order to improve the delivery of the enzyme and internalization to the desired cells, as well as to achieve the combined effect of the action of two molecules. Nevertheless, authors mention a number of delivery vehicles (liposomes, chitosan, MnO2, microemulsion, iron-tannic acid nanocapsules, etc.). Thereunder, I suggest to change the manuscript title to reflect the content better.

The manuscript looks more like listing possible enzymes classes and constructs based on them that influence the cancer cell metabolism. The authors should add more analysis of the use of one or another toxin to affect specific cells, there may be a comparison of the effectiveness of enzymes with each other.

Reviewer 2 Report

Authors should high lights on

  1. Conception, novel aspects and evolution of patented technologies reported for  Advanced delivery systems for enzyme drugs in cancer therapy.
  2. Features, limitations and advantages of the Advanced delivery systems for enzyme drugs in cancer therapy from in vivo and clinical studies (Table format ).
  3. Formulation consideration and  drug carrier systems to deliver  enzyme drugs in cancer therapy (Table format ).
  4. Generalized Risk Estimation Matrix (REM) for enzyme drugs( Low: parameter with low risk; Medium: parameter with medium risk; High: parameter with high risk)
  5. Advanced technologies potentially applicable in personalized treatments mentioned in recent clinical trials.

Round 2

Reviewer 2 Report

I accept.